# Escin’s Multifaceted Therapeutic Profile in Treatment and Post-Treatment of Various Cancers: A Comprehensive Review

**DOI:** 10.3390/biom13020315

**Published:** 2023-02-07

**Authors:** Sunnatullo Fazliev, Khurshid Tursunov, Jamoliddin Razzokov, Avez Sharipov

**Affiliations:** 1Max Planck School Matter to Life, Jahnstrasse 29, 69120 Heidelberg, Germany; 2Faculty of Engineering Sciences, Heidelberg University, Im Neuenheimer Feld 205, 69120 Heidelberg, Germany; 3Department of Inorganic, Physical and Colloidal Chemistry, Tashkent Pharmaceutical Institute, Oybek Street 45, Tashkent 100015, Uzbekistan; 4State Center for Expertise and Standardization of Medicines, Medical Devices and Medical Equipment, Agency for the Development of the Pharmaceutical Industry under the Ministry of Health of the Republic of Uzbekistan, Ozod Street 16, Tashkent 100002, Uzbekistan; 5Institute of Fundamental and Applied Research, National Research University TIIAME, Kori Niyoziy 39, Tashkent 100000, Uzbekistan; 6College of Engineering, Akfa University, Milliy Bog Street 264, Tashkent 111221, Uzbekistan; 7Department of Physics, National University of Uzbekistan, Universitet 4, Tashkent 100174, Uzbekistan; 8Laboratory of Experimental Biophysics, Centre for Advanced Technologies, Universitet 7, Tashkent 100174, Uzbekistan; 9Department of Analytical and Pharmaceutical Chemistry, Institute of Pharmaceutical Education and Research, Yunusota Street 46, Tashkent 100114, Uzbekistan

**Keywords:** angiogenesis, apoptosis, cancer, escin, inflammation, metastasis, saponin

## Abstract

Although modern medicine is advancing at an unprecedented rate, basic challenges in cancer treatment and drug resistance remain. Exploiting natural-product-based drugs is a strategy that has been proven over time to provide diverse and efficient approaches in patient care during treatment and post-treatment periods of various diseases, including cancer. Escin—a plant-derived triterpenoid saponin—is one example of natural products with a broad therapeutic scope. Initially, escin was proven to manifest potent anti-inflammatory and anti-oedematous effects. However, in the last two decades, other novel activities of escin relevant to cancer treatment have been reported. Recent studies demonstrated escin’s efficacy in compositions with other approved drugs to accomplish synergy and increased bioavailability to broaden their apoptotic, anti-metastasis, and anti-angiogenetic effects. Here, we comprehensively discuss and present an overview of escin’s chemistry and bioavailability, and highlight its biological activities against various cancer types. We conclude the review by presenting possible future directions of research involving escin for medical and pharmaceutical applications as well as for basic research.

## 1. Introduction

Advances in medicine and pharmacology allow us to diagnose, prevent and treat many diseases that hamper the quality of life. These achievements are partially realised thanks to developments in molecular biology, genetics and drug discovery. However, several basic challenges, like drug resistance and cancer treatment, continue to pose a predicament. Although drug design is progressing through computational, machine learning and automation methods [1,2], natural products remain a cornerstone in pharmaceutical science. Metabolites of microorganisms, plants and animals are rich sources of valuable chemicals that demonstrate potent activities against a broad range of diseases. Cancer, as a bottleneck of medicine, is a leading cause of death without differentiating between wealthy and poor countries. Worldwide, it takes the lives of approximately 10 million people per annum [3]. This disease is characterized by abnormal growth of cells and results in very serious complications, poor quality of life and shortened life expectancy. Cancer diagnosis and treatment are expensive and difficult, and not always possible. Medicinal plants offer cheap, readily available sources of active pharmaceutical compounds (APCs) against many diseases. Approximately 25% of all the newly approved FDA drugs between 1981 and 2019 are considered to be natural products or derived from natural products [4]. In this time frame, FDA approved 182 new drugs against cancer, 120 of which were natural product-based drugs [4]. It is remarkable how APCs of plant origin can be used to combat serious diseases like cancer in a variety of ways: ranging from diagnosis to treatment to post-treatment care.

One group of multifaceted APCs of plant origin is escin. These are triterpenoid saponins usually isolated from the seeds of Horse chestnut trees. As we shall show below, more than 30 escin isomers have been isolated and studied. Originally, escin was proven to be efficient against inflammation, chronic venous insufficiency and gastroprotection. But over the past two decades, numerous novel activities of escin have been published, such as anti-oxidative and anti-cancer effects. These studies demonstrated that escin has anti-proliferative and anti-metastasis effects against more than 15 different cancer types. In addition, a new trend—composite drugs of escin and other APCs—is on the rise. Therefore, escin’s broad scope of biological activities combined with various composite drugs makes it appealing for use in cancer treatment and post-treatment care.

Although there are reviews of escin beginning in the 2000s, all of these reviews focus on the most common biological activities of escin; few have presented the chemistry and bioavailability of escin. Therefore, we sought to comprehensively review the chemistry, pharmacology and pharmaceutics of escin by extensively discussing and presenting a big picture of escin’s role in the treatment of cancer, as well as providing an updated resource with recent findings.

In literature, two types of escin, α- and β-escin, are often reported and they correspond to **escin Ia** and **escin Ib**, respectively [5]. Pharmacologically relevant escin is β-escin, which is less soluble in water. There are some disparities in the nomenclatural use of the word “escin”. Generally, the terms aescin and escin are equivalent. However, there are literature sources that refer to these terms differently; some refer to it as a mixture of 2, 4 [6,7] or many saponins [8,9,10,11,12]. The European Medicines Agency defines escin as a mixture of saponins with M_r_ = 1131 g/mol [13]. This corresponds to a mixture of **escin Ia**, **Ib**, **IVc**, **IVd** and **VIb** and **isoescin Ia**, **Ib** and **VIIa**. These discrepancies could be the result of various isolation methods that influence the composition of the final extracted product. In the context of pharmacology and pharmaceutics, we advocate using the terms “aescin” or “escin” to refer to a mixture of many saponins (the exact number is not defined; we shall show 34 escin isomers in Table 1) whose aglycone substructure is either that of protoascigenin or barrigenol C. In the course of this review, we shall use this nomenclature.

## 2. Chemistry of Escin

The chemical structure of escin is made up of two parts: (i) the aglycone part as a derivative of protoascigenin and (ii) the glycone part. β-d-glucouronopyranosyl acid, β-d-glucopyranose, β-d-galactopyranose and β-d-xylopyranose constitute the glycone part of escin (Figure 1).

The aglycone substructure of escin has three or four OH groups at C21, C22, C23 and C28 and thus can produce many isomers by forming various esters with acetic, angelic and tiglic acids (Table 1). The OH group at C3 escin is connected via a glycoside bond to β-d-glucouronopyranosyl acid, which in turn is connected to a β-d-glucopyranosyl unit (1→4). These two sugars are present in all escins. The third sugar glycosylates the β-D-glucouronopyranosyl acid and, depending on the escin isomer, can be one of the following moieties: β-d-glucopyranosyl (1→2), β-d-galactopyranosyl (1→2) and β-d-xylopyranosyl (1→2) (Table 1).

Isolated escin is colourless fine crystals or white powder. Typical melting temperatures of 187–230 °C are observed. The aglycone part of escin is relatively stable; even at temperatures as high as 150 °C, the saponins do not undergo denaturation. However, under certain conditions, deacetylation might occur by producing deacetylescin (Table 1). For example, alkaline hydrolysis of escin liberates acetic, angelic and/or tiglic acids resulting in the respective deacetylescin. Further methanolysis of deacetylescin breaks the glycoside bond, and the respective aglycones without acidic moieties are formed. Escin isomers can be differentiated by determining melting points and infrared, nuclear magnetic resonance and mass spectra.

## 3. Pharmacokinetics and Bioavailability of Escin

The bioavailability of APCs is closely related to their structure, absorption in the body, passage through barriers and metabolism. The pharmacokinetics and bioavailability of escin have been studied extensively. Due to the fact that escin was known in traditional medicine and has been used for hundreds of years, early reports of isolation, discovery and structural studies were published concurrently with pharmacokinetic studies. Scientists were eager to identify the essential pharmacological components of escin, which is a complex mixture of saponins. As mentioned in earlier sections, different isomers of escin demonstrate unique biological activities. Pharmacokinetic studies also shed light on distinct behaviours of escin isomers in terms of bioavailability and kinetics.

Wu and colleagues systematically studied the pharmacokinetics of various escin isomers in animals and were able to obtain valuable information about the relationship between the structure and pharmacokinetic properties of the isomers. It was shown that **escin Ib** and **isoescin Ib**, after intake, were converted into **escin Ia** and **isoescin Ia**, respectively [20]. Based on this phenomenon, they concluded that escin isomers with a tigeloyl moiety (present in **escin Ia** and **isoescin Ia**) are better able to permeate into the intracellular space than escin isomers with the angeloyl moiety (present in **escin Ib** and **isoescin Ib**, also see Table 1). They also found that preparations containing several escin isomers have a longer duration of action than single, pure isomers of escin [21].

Several studies demonstrated very high bioavailability (>90% compared to reference substances) of escin irrespective of the form of medicine, i.e., extracts or tablets [22,23]. Nonetheless, there are assumptions that circadian rhythms and food can affect escin’s bioavailability [22,23]. Escin reaches its maximal plasma concentration about 2 h after the first dose and circulates in blood with a half-life of t_1/2_ = 6–8 h. Multiple dose studies (up to a week) with escin tablets showed that its pharmacokinetics does not change over time [24]. Some soft dosage forms of escin, especially hydrophilic gels, showed enhanced bioavailability and prolonged release. For instance, acetylated, annealed starch [25] and polyacrylate cross-linked hydrophilic polymer [26] positively contributed to the prolonged release of escin from such gels. On the contrary, it has been observed that a hydrogel made of methylcellulose, which is hydrophobic polymer, released escin too quickly [25]. Another emerging drug delivery approach—encapsulating drugs into liposomes—has already been applied to escin, and the results demonstrated that such liposomes increased escin’s bioavailability several times [27].

## 4. Escin’s Relevant Biological Activities in Cancer Therapy

### 4.1. Anti-Cancer Effects

It has long been known that escin has potential anti-cancer effects. Escin’s anti-cancer mechanisms can be grouped into three categories: (i) inducing apoptosis, (ii) reducing cell proliferation and (iii) inhibiting metastasis. Over the past two decades, the results of numerous in vitro and in vivo studies revealed the efficacy of escin in suppressing and/or preventing over 15 cancer types, which are summarised in Table 2.

Regarding the molecular mechanisms of escin-induced apoptosis of cancer cells, ample evidence has been compiled to support the mitochondria-mediated (intrinsic) apoptosis pathway and reactive oxygen species (ROS)-induced DNA damage (Figure 2). Nevertheless, there are reports of death-receptor-induced (extrinsic) apoptosis induced by escin [29,48].

The intrinsic apoptosis pathway is a mitochondria-actuated event at which various stimuli decrease membrane potential and hence increase the permeability of the mitochondrial membrane [60]. Consequently, several enzymes and molecules are released from mitochondria to activate the caspase-dependent mitochondrial apoptosis. It is believed that escin disrupts several pathways leading to the mitochondrial depolarisation. This includes the inhibition of NF–κB, which results in the downregulation of anti-apoptotic Bcl family proteins [55,58,61] and ROS/p38 MAPK pathway [52] (Figure 2). Elevated caspase protein levels corroborate this mechanistic explanation (Table 2). Meanwhile, ROS can give rise to cancer cell death through DNA damage [32,37,38].

Molecular mechanisms of the anti-proliferative effects of escin mostly depend upon the inhibition of the NF–κB, JAK/STAT and ERK1/2 pathways. Here, the NF–κB pathway, which is the central target of escin’s anti-inflammatory effects, is believed to be one of the key linkages between inflammation and cancer [62]. Apart from mediating the proinflammatory [63] and anti-apoptotic [61] genes, NF–κB also targets the genes of cell proliferation regulators such as cyclins and cytokines [64] (Figure 2). Moreover, NF–κB contributes to cancer metastasis and invasion through the upregulation of adhesion molecules [64] and chemokines [47,57]. Therefore, NF–κB inhibition is the main contribution to the anti-cancer, as well as anti-inflammatory effects of escin. In addition, ERK1/2, JAK/STAT and Akt proteins and their signalling pathways were also shown to be affected by escin to inhibit cell proliferation. At a cellular level and depending on cancer type, escin-treated cancer cells were caught in cell cycle arrests at the G0/G1, G1/S and G2/M cell cycle phases (Table 2).

Anti-metastasis effects of escin, albeit being less investigated, seem to encompass a wide variety of mechanisms. Many reported the downregulation of NF–κB and downstream molecules of NF–κB signalling pathways, like vascular epidermal growth factor (VEGF), IL-8 [57], TNF [47] and Bcl family proteins [6]. Others advocated contributions from Akt and ERK1/2 [41,51]. Meanwhile, more recent studies communicated that escin inhibits metastasis by regulating the tumour microenvironment, for instance, by blocking ECM production, inhibiting hypoxia-inducible factor 1-alpha (HIF1α) targeted protein expression [54] and downregulating inducible nitric oxide synthase (iNOS) [48], RhoA and Rock proteins [49]. Escin’s regulation of the tumour microenvironment is evidenced by the downregulation of several matrix metalloproteinases (MMPs) [31,65]. Escin also displays potent anti-angiogenetic effects to prevent tumour metastasis, which will be discussed next.

### 4.2. Anti-Angiogenetic Effects

Escin’s anti-angiogenetic effects are also of interest, especially in the context of cancer treatment. Several in vitro and in vivo studies have indicated escin’s efficacy in the inhibition of tumour invasion, migration and metastasis (Table 2). It was demonstrated that escin inhibits proliferation [66] and migration of HUVECs [67]. The exact mechanisms by which escin alters HUVECs proliferation and migration remain elusive, but it was suggested that escin acts on HUVECs through Akt, p38/MAPK and ERK signalling pathways and by inhibition of PKC-α, EFNB2 and growth factor expression [66,67]. Escin’s anti-angiogenic effects are also facilitated by disrupting the extracellular matrix (ECM) modelling and adhesion. Reports showed escin to inhibit the secretion of vascular epidermal growth factor (VEGF) in HUVECs [31], matrix metalloproteinases (MMP-3 and MMP-9) in rats [68] and ECM production in mesothelial cells and fibroblasts [54]. Because of the anti-inflammatory, anti-oedematous and venotonic effects, scientists were quick to consider escin for the treatment of widespread chronic venous insufficiency (CVI). Escin’s efficacy against CVI has been tested by numerous in vitro, in vivo and clinical trials [69,70]. Nowadays, standardised horse-chestnut dry extract is used at various stages of CVI, varicose and other venous disorders associated with oedema [13].

### 4.3. Anti-Inflammatory Effects

Inflammation is a basic immune response to infection and injury that sets complex defence signalling pathways on “fire”. When out of control, inflammation can be chronic, which is also associated with tumour development [64]. Escin’s potential effect in reducing inflammation was already shown in the 1960s [71]. Extensive research on this topic proposed several key players of inflammatory networks as the targets of escin’s anti-inflammatory properties. There are numerous reports indicating that escin supresses the activation of nuclear factor kappa B (NF–κB) and inflammation processes initiated by this protein. This is substantiated by in vitro and in vivo studies where escin decreases the level of proinflammatory cytokines [72,73,74,75,76,77], tumour necrosis factor α (TNF-α), and interleukins IL-1β and IL-6, whose production is directly regulated by NF–κB as a transcription factor. Results of many studies with proinflammatory factors (TNFα, IL-1β), Toll-receptor ligands (lipopolysaccharides) and 11-β-hydroxysteroid dehydrogenase type 2 (11β-HSD2) put forward a glucocorticoid-like (GC) mechanism of escin in the regulation of inflammation (Figure 3).

Initially, it was shown that escin upregulates adrenocorticotropin and corticosterone levels [78]. Hence, anti-inflammatory effects of escin were accredited to increased levels of GCs [7]. However, following results indicated that escin’s anti-inflammatory effects are realised through upregulation of glucocorticoid receptor (GR) and induction of the conformation change in the receptor to facilitate GC binding [75,77,79,80]. This conformational change is necessary to expose GR to nucleoporins and importins that transfer the GC-bound GR into the nucleus. After, the GR can be recruited to indirect binding sites in DNA by interacting with DNA-TF complexes [81]. These interactions between GR and DNA-TF complexes directly influence gene transcription by changing DNA–TF interactions and/or by recruiting other TFs and cofactors that can result in suppression of proinflammatory factors like TNFα, IL-1β, and IL-6 (Figure 3).

Meanwhile, GC-bound GRs can inhibit the activation of NF–κB in the cytosol. Studies indicate that escin itself can inhibit the activation of NF–κB by downregulating the expression of p65 [57,75] and by inhibiting the IKK complex (comprised of the IKKα and IKKβ catalytic subunits and the IKKγ regulatory subunit) activation [47]. P65 is a necessary component of the p50/p65 dimer that is a primary target of the canonical NF–κB activation pathway [64]. There are other findings that report the inhibition of NF–κB by escin in mice [82,83], rats [75,84], human bladder cancer cells [28], human endothelial cells [85], and human pancreatic cancer cells (BxPC-3, PANC-1) [55,57]. However, the exact mechanisms of direct NF–κB inhibition by escin still remain unresolved. Decreased TNF-α, IL-1β, IL-6, and NO levels and increased antioxidant factors are ascribed as plausible explanations for escin-induced NF–κB inhibition [7,72,74,86].

### 4.4. Antioxidant, Protective and Ameliorative Effects

In general, escin’s potent anti-inflammatory and antioxidant activities prompted its evaluation for alleviating pain and complications in several diseases and might be very useful in chemotherapy and surgical treatment of cancer diseases. This is very crucial for healthy cells to survive detrimental conditions of cancer chemotherapies and act as a scavenger of ROS whose levels are elevated in tumour microenvironments. For instance, escin increased glutathione, catalase and SOD activities, decreased MMP-9 levels and mitigated cardiac autonomic neuropathy in rats with diabetes induced by streptozotocin [87]. The same antioxidant effects of escin (increased catalase and SOD activities) appear to be useful against cyclophosphamide-induced cardiotoxicity [88]. Escin’s antioxidant mechanisms can also be realised via the AKT-Nrf2 signalling pathway, as shown in studies of H_2_O_2_-induced cytotoxicity in established retinal pigment epithelium (ARPE-19) and primary murine RPE cells [89].

Escin also displays promising effects in brain injuries and neurodegenerative diseases - a hot topic of contemporary medicine. Escin attenuated cerebral ischemia-reperfusion injury in rats by reducing the volume of cerebral infarct and water content, and by ameliorating the neurological deficit [90]. Likewise, escin was effective in inhibiting inflammation, attenuating cognitive deficit and protecting hippocampal neurons in ischemic brain injury [91]. There is a growing interest in escin’s potential role in neurodegenerative diseases. I is reported that oral treatment with escin diminished behavioural impairments, oxidative stress and inflammation in the chronic MPTP/probenecid mouse model of Parkinson’s disease (PD) by reducing the levels of TNF-α, IL-6, IL-4, IL-10, SOD and catalase [86]. Later, escin’s positive effects on mitochondrial dysfunction, oxidative stress and apoptosis in a mouse PD model was demonstrated [92]. Meanwhile, escin was proven to be an autophagy inducer that degrades mutant huntingtin protein (mHtt) and inhibits mHtt-induced apoptosis in vitro and in vivo [93]. Escin’s autophagy induction mechanisms in HT22 cells were attributed to the regulation of mTOR and ERK pathways [93].

Ameliorative effects of escin also rely on intervening inflammatory processes, mostly through the NF–κB pathway. For instance, Zhang et al. recently reported the ameliorative effects of escin on neuropathic pain in chronic constriction injury of the sciatic nerve by suppressing the NF–κB pathway and its targets: pro-inflammatory cytokines, TNF-α, glial fibrillary acidic protein and nerve growth factor [94]. Escin was found to be a useful analgesic in bone cancer pain by suppressing inflammation and microglial activation, possibly through the suppression of the p38 MAPK/c-Fos signalling pathway [95]. Moreover, escin can contribute to muscle regeneration and prevention of muscle atrophy by reducing inflammatory infiltration, fibrosis and by increasing the number of muscle fibres [96,97].

## 5. Conclusions and Future Perspectives

In this paper, we provided a comprehensive discussion of escin’s chemistry, structural characteristics, and its multifaceted role in cancer therapy. Escin’s chemistry and physicochemical properties were presented, including structural information on the 34 isomers. We showed that there is ample evidence to suggest the anti-cancer effects of escin to be mainly realised through the NF–κB pathway and can be summarised into three groups: apoptotic, anti-cell proliferative and anti-metastasis effects. Escin’s broad biological activity scope enables it to be used as a potential anti-cancer compound, a supplementary medicine in cancer therapy and as an adjunct compound in composite drugs. In addition, escin can be used in treatment and post-treatment periods of cancer therapy due to its anti-inflammatory, anti-oxidative, protective and ameliorative effects. We reviewed these effects of escin and concluded that escin’s anti-inflammatory effects are realised by the GC-like mechanism.

Escin has been an interesting and wide-reaching topic in the pharmaceutical industry. Hence, it is a subject of systematic research. Still, some critical research is needed to further advance the state of the art. For instance, pharmacological studies of escin were performed in vitro and in vivo; however, as technology progresses, there are now new methods and techniques that can offer more precise analysis. One such trend is single-cell studies. As the mechanisms of escin’s pharmacological activities are not yet fully realised, single-cell experiments can provide a platform to elucidate mechanisms of action. This is especially true for the biological activities of escin against cancer cells. Computational advances allow us to work with big data, and high-throughput single-cell experiments are becoming popular. Therefore, it might be of significant interest to reveal the finest mechanistic details of escin’s biological activities in the context of cancer. This should enable us to understand the nature of cancer diseases better and to increase the therapeutic efficacy of medicines, including escin against such diseases.

Another important area of research is to ensure the protection of our natural resources. The development of new medicines based on escin’s structure can contribute to “green pharmacy”. Apart from escin’s use as an APC, it can be used to develop drug carriers that are cheap, safe and with additional therapeutic effects. Modification of escin with synthetic or other biological surfactants is also an interesting direction. This could be an alternative way to design semi-synthetic surfactants with improved properties, which are crucial to producing drug delivery tools that increase the solubility, bioavailability and stability of drugs, especially those with high molecular weights. Such nanoparticles and vesicles containing escin can enhance drug action by increasing the membrane’s permeability and sensitising cancer cells to chemotherapy drugs, as well as by contributing to therapeutic activity via its anti-inflammatory activity.

## Figures and Tables

**Figure 1 biomolecules-13-00315-f001:**
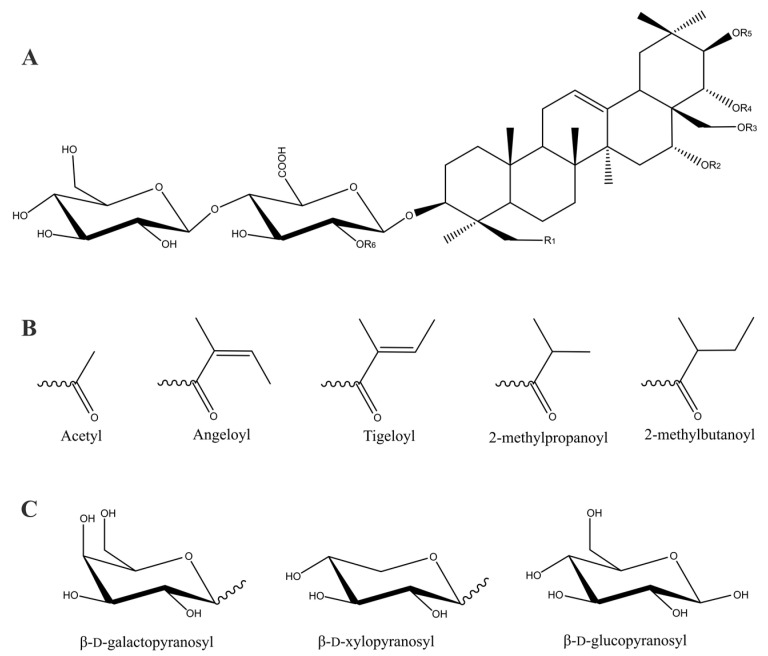
General chemical structure of escin (**A**), structures of various acid moieties (**B**) and sugar units (**C**) found in escin.

**Figure 2 biomolecules-13-00315-f002:**
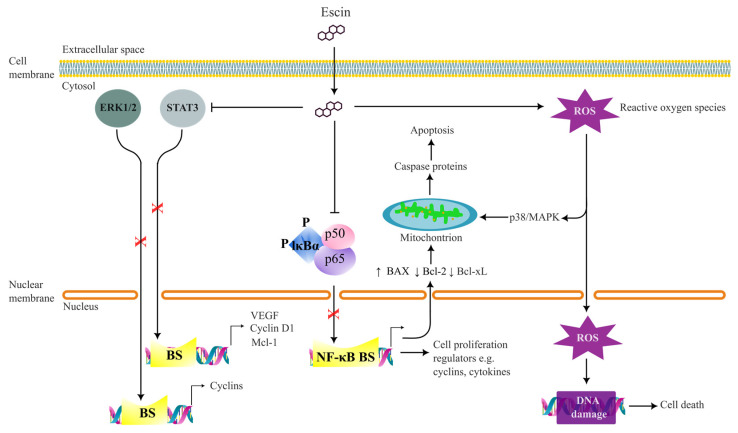
Possible molecular targets of escin’s anti-cancer mechanisms. Abbreviations: BS—binding site, NF-κB—Nuclear factor kappa B, ERK1/2—extracellular signal-regulated kinase 1/2, Bcl-2—B-cell lymphoma 2 proteins, BAX—BCL2-associated X, Bcl-xL—X-linked inhibitor of apoptosis protein (xIAP), VEGF—vascular epidermal growth factor, MAPKs—p38 mitogen-activated protein kinases, IκBα—cytoplasmic inhibitory protein of NF-κB, STAT3—signal transducer and activator of transcription protein 3.

**Figure 3 biomolecules-13-00315-f003:**
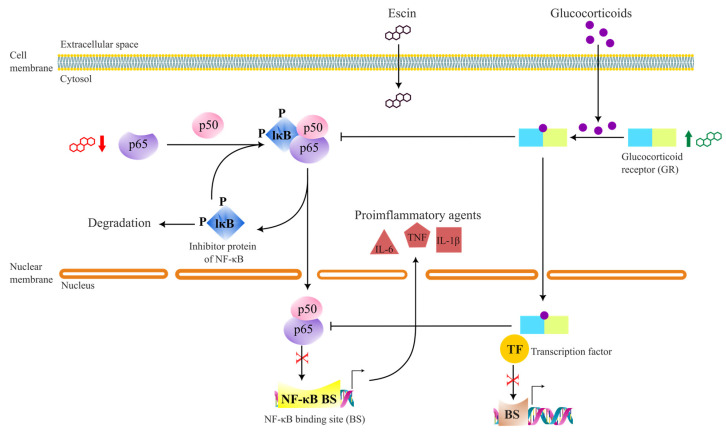
Overview of anti-inflammatory mechanisms of escin at the molecular level.

**Table 1 biomolecules-13-00315-t001:** Chemical structures of escin. (See also Figure 1).

Compound	R_1_	R_2_	R_3_	R_4_	R_5_	R_6_	Ref.
**Escin Ia**	OH	H	H	Acetyl	Tigeloyl	β-d-glucopyranosyl	[8,12]
**Escin Ib**	OH	H	H	Acetyl	Angeloyl	β-d-glucopyranosyl	[8,12]
**Escin IIa**	OH	H	H	Acetyl	Tigeloyl	β-d-xylopyranosyl	[8]
**Escin IIb**	OH	H	H	Acetyl	Angeloyl	β-d-xylopyranosyl	[8]
**Escin IIIa**	H	H	H	Acetyl	Tigeloyl	β-d-galactopyranosyl	[8,9]
**Escin IIIb**	H	H	H	Acetyl	Angeloyl	β-d-galactopyranosyl	[11]
**Escin IV**	OH	H	H	Acetyl	Acetyl	β-d-glucopyranosyl	[11]
**Escin IVc**	OH	H	Acetyl	Tigeloyl	H	β-d-glucopyranosyl	[14]
**Escin IVd**	OH	H	Acetyl	Angeloyl	H	β-d-glucopyranosyl	[14]
**Escin IVe**	OH	H	Tigeloyl	H	H	β-d-glucopyranosyl	[14]
**Escin IVf**	OH	H	Angeloyl	H	H	β-d-glucopyranosyl	[14]
**Escin IVg**	OH	H	H	Tigeloyl	H	β-d-glucopyranosyl	[15]
**Escin IVh**	OH	H	H	Angeloyl	H	β-d-glucopyranosyl	[15]
**Escin V**	OH	H	H	Acetyl	2-methylpropanoyl	β-d-glucopyranosyl	[11]
**Escin VI**	OH	H	H	Acetyl	2-methylbutanoyl	β-d-glucopyranosyl	[11]
**Escin VIb**	OH	Angeloyl	H	H	Acetyl	β-d-glucopyranosyl	[15]
**Isoescin Ia**	OH	H	Acetyl	H	Tigeloyl	β-d-glucopyranosyl	[11]
**Isoescin Ib**	OH	H	Acetyl	H	Angeloyl	β-d-glucopyranosyl	[11]
**Isoescin IIa**	OH	H	Acetyl	H	Tigeloyl	β-d-xylopyranosyl	[16]
**Isoescin IIb**	OH	H	Acetyl	H	Angeloyl	β-d-xylopyranosyl	[16]
**Isoescin IIIa**	H	H	Acetyl	H	Tigeloyl	β-d-galactopyranosyl	[16]
**Isoescin IIIb**	H	H	Acetyl	H	Angeloyl	β-d-galactopyranosyl	[16]
**Isoescin V**	OH	H	Acetyl	H	2-methylpropanoyl	β-d-glucopyranosyl	[11]
**Isoescin VIa**	OH	H	Acetyl	H	2-methylbutanoyl	β-d-glucopyranosyl	[17]
**Isoescin VIIa**	OH	H	Acetyl	H	Tigeloyl	β-d-galactopyranosyl	[17]
**Isoescin VIIIa**	H	H	Acetyl	H	Angeloyl	β-d-glucopyranosyl	[17]
**Deacylescin I**	OH	H	H	H	H	β-d-glucopyranosyl	[8,18]
**Deacylescin Ia**	OH	H	H	H	Tigeloyl	β-d-glucopyranosyl	[19]
**Deacylescin Ib**	OH	H	H	H	Angeloyl	β-d-glucopyranosyl	[19]
**Deacylescin II**	OH	H	H	H	H	β-d-xylopyranosyl	[10,18]
**Deacylescin IIa**	OH	H	H	H	Tigeloyl	β-d-xylopyranosyl	[19]
**Deacylescin IIb**	OH	H	H	H	Angeloyl	β-d-xylopyranosyl	[19]
**Deacylescin III**	H	H	H	H	H	β-d-galactopyranosyl	[8]

**Table 2 biomolecules-13-00315-t002:** Anti-cancer effects of escin revealed by in vitro or in vivo studies.

Cancer Type	Model System	Anti-Cancer Effects and Mechanisms of Escin by			Ref.
		Inducing Cell Apoptosis	Decreasing Cell Proliferation	Inhibiting Metastasis and Invasion	
Bladder cancer	T24, J82, TCCSUP, and RT-4 cell lines	Through death receptor and mitochondria mediated pathways	↓ NF–κB p65, partially affects STAT3 expression		[28]
Breast cancer	MDA-MB-231 cell line			↓ LOXL2, ↓ c-Myc, ↓ GSL1, ↑ASCT2	[29,30]
	Mice and MCF-7 cell line	Yes	Yes	↑ p53, ↓ Bcl-2	[31,32,33,34]
Colon cancer	HT-29 cell line		Cell cycle arrest at G1/S phase mediated by induction of p21^WAF1/CIP1^		[35]
	LoVo cell line		Yes		[36]
Colorectal cancer	Mice and HCT116, HCT8 cell lines	Through DNA damage	Yes		[37]
	Mice and HCT116, HCT8 cell lines	Through DNA damage ↑ TIGAR ↑ ROS	Yes		[38]
Cholangiocarcinoma	QBC939, MZ-ChA-1 Sk-ChA-1 cell lines	↑ caspase-3, ↓ Bcl-2	Cell cycle arrest at G1 and G2/M phases		[39]
	QBC939 cell line	Sensitised to 5-FU and VCR	↓ P-glycoprotein, ↓ GSK3b/ catenin		[40]
Gastric cancer	AGS cell lines			Through Akt signalling pathway	[41]
Glioblastoma multiforme	Classical and mesenchymal glioblastoma-initiating cells	Through mitochondria-mediated pathway	Yes		[42]
Hepatocellular carcinoma	SMMC-7721 cell line	↑ caspases 3, 8, 9; ↓ Bcl-2 with 5-FU	Cell cycle arrest at G0/G1 with 5-FU		[43]
	HepG2 cell line	↑ PARP, AIF, BAX, and Bcl-2	Cell cycle arrest at G1/S phase		[44]
	HepG2 cell line		↓ Akt/JAK/STAT, ↓ cyclin D1, ↓ Bcl-2, ↓ Bcl-xL, ↓ survivin, ↓ Mcl-1, ↓ VEGF; sensitised to Dx, PTX		[45]
Leukaemia	Jurkat T-cell line	↑ Caspases-3, 8, 9, ↓ PARP, ↓ Bcl-2, ↑ ROS	Yes		[32,46]
	CEM cell line		Yes		[32]
	KBM-5 cell line	TNF-induced apoptosis	↓ TNF ↓ NF–κB	↓ TNF ↓ NF–κB	[47]
Lung cancer	A549 cell line		Through JAK/STAT signalling pathway	↓ iNOS	[48]
	Mice and H460 cell line		↓ ALDH1A1 ↓ p-Akt, ↑ p21	↓ RhoA and Rock	[49]
	A549 cell line	↑ BAX, ↑ caspase-3	Cell cycle arrest at G0/G1phase		[50]
Melanoma	SK-MEL5 and B16F10 cell lines		↓ NF–κB, ↓ IκB	Through ERK1/2 signalling	[51]
Osteosarcoma	Mice and MNNG/HOS, Saos-2, MG-63, U-2OS, HUVEC cell lines	Through ROS/p38 MAPK signalling pathway			[52]
	Mice and MG-63, OS732 cell lines	↑ Caspases-3, 8, 9	↓ PI3K/Akt pathway		[53]
Ovarian cancer	HeyA8, SNU-119, Kuramochi, Ovcar4, and Ovcar5 cell lines			↓ Autophagy-dependent CSC differentiation, ↓ Stromal ECM production driven by HIF1α	[54]
Pancreatic cancer	Mice and BxPC-3, PANC-1 cell lines	↓ NF–κB, ↓ c-Myc, ↓ COX-2, ↓ cyclin D1, ↓ survivin, ↓ Bcl-xL, ↓ Bcl-2, ↑ caspase-3			[55]
	COLO357, MIA-Paca, Panc-1, cell lines	Yes	↓ NF–κB, ↓ cyclin D, sensitised cells to cisplatin		[56]
	BxPC-3, AsPC-1, SW1990 cell lines		↓ NF-κB	↓ IL-8, ↓ VEGF	[57]
Prostate cancer	Mice and CRPC, PC-3, DU-145 cell lines	↑ c-caspase-3, ↑ BAX, ↓ Bcl-2, ↓ cIAP-1, ↓ cIAP-2 ↓ xIAP, ↑ PARP	Cell cycle arrest at G2/M-phase		[58]
Renal cancer	786-O and Caki-1 cell lines	↓ Bcl-2, ↑ ROS	Cell cycle arrest at G2/M arrest		[59]

Abbreviations: Dx—doxorubicin, IL—interleukin, PTX—paclitaxel, VCR—Vincristine, LOXL2—lysyl oxidase-like 2, EMT—epithelial-mesenchymal transition, 5-FU—5-Fluorouracil, CSC—cancer stem cell, GLS1—kidney type glutaminase, ASCT2—alanine-serine-cysteine 2 protein, MAPKs—p38 mitogen-activated protein kinases, NF-κB—Nuclear factor kappa B, ROS—reactive oxygen species, TIGAR—TP53-induced glycolysis and apoptosis regulator, CXCL16—chemokine (C-X-C motif) ligand, iNOS—inducible nitric oxide synthase, ERK1/2—extracellular signal-regulated kinase 1/2, PARP (c)—poly (adenosine diphosphateribose) polymerase, Bcl-2—B-cell lymphoma 2 proteins, BAX—BCL2-associated X, Bcl-xL—X-linked inhibitor of apoptosis protein (xIAP), ALDH—aldehyde dehydrogenase, p-Akt—phospho-Akt, HUVECs—human umbilical vein endothelial cells, VEGF—vascular epidermal growth factor, ECM—extracellular matrix, and HIF1α—Hypoxia-inducible factor 1-alpha.

## Data Availability

Not applicable.

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
