# Peer review of "Escin’s Multifaceted Therapeutic Profile in Treatment and Post-Treatment of Various Cancers: A Comprehensive Review"

_biomolecules, 2023, doi:10.3390/biom13020315_

Round 1

Reviewer 1 Report

The review article by Fazliev et al. “Escin’s multifaceted roles in the treatment of cancer diseases and beyond: a comprehensive review,” comprehensively presents an overview of chemistry, the bioavailability of escin, and its biological activities against various cancer types. The article begins with an introduction section where an overview of the escins is given and highlights their types and nomenclature. This is followed by sections that discuss in detail the chemistry, pharmacokinetics, bioavailability, and biological activity against several cancers. In addition, the anti-inflammatory, anti-angiogenic, and anti-oxidant effects of escins are also discussed. Following this is the conclusion and future perspective section, which is very interesting as the authors have provided their thoughts about improving future research on escins. Three figures and two tables suffice the content of this review. The art and graphics in the figures are appropriate, and the details on the figures are well-highlighted. Overall, the review article is very interesting, with up-to-date literature.

Minor revision:

1.    The numbering of a few sections needs to be changed for consistency. 3.3. should be 4.2 and so on.

2.    The authors should explain how this review has improved from the previously published article on the same topic. (Molecular targets and anti-cancer potential of escin.  Cancer Lett,  2018 May 28;422:1-8).

3.    The paper needs minor editing for English and grammatical errors.

Author Response

Prof. Dr. Avez Sharipov                       

Title, function or space                                                                         Biomolecules MDPI

Faculty of Pharmacy,                                                                                           

Department of Inorganic,                                                                     Prof. Dr. Vladimir N. Uversky 

Physical and Colloidal Chemistry                                                        Editor-in-Chief of 

Tashkent Pharmaceutical Institute                                                

Oybek Street 45                                                                                                                                 

100015 Tashkent, Uzbekistan

DATE                                                                  SUBJECT

30 January 2023                          Manuscript submission

Dear Editor,

Dear Editor,

We would like to thank you and the referees for the evaluation of our manuscript submitted to Biomolecules (Manuscript ID:2109114 ) by Sunnatullo Fazliev, Khurshid Tursunov, Jamoliddin Razzokov and Avez Sharipov, entitled “Escin’s multifaceted roles in the treatment of cancer diseases and beyond: a comprehensive review”.

We read all three referees’ reports carefully. The referees recommended our paper for publication after considering some comments. Indeed, referees’ comments were helpful to increase the value of our manuscript. Please find below our answers to the referees’ comments, as well as the corresponding changes and improvements made in the manuscript.

Sincerely,

Dr. Jamoliddin Razzokov

Reply to the Referees

We would like to thank referees for the useful comments/suggestions that, indeed, are valuable and improve the quality of our manuscript. We have taken all comments into consideration and revised the manuscript accordingly. All additions and modifications to the manuscript are formatted in red text.

Reviewer 1:

The review article by Fazliev et al. “Escin’s multifaceted roles in the treatment of cancer diseases and beyond: a comprehensive review,” comprehensively presents an overview of chemistry, the bioavailability of escin, and its biological activities against various cancer types. The article begins with an introduction section where an overview of the escins is given and highlights their types and nomenclature. This is followed by sections that discuss in detail the chemistry, pharmacokinetics, bioavailability, and biological activity against several cancers. In addition, the anti-inflammatory, anti-angiogenic, and anti-oxidant effects of escins are also discussed. Following this is the conclusion and future perspective section, which is very interesting as the authors have provided their thoughts about improving future research on escins. Three figures and two tables suffice the content of this review. The art and graphics in the figures are appropriate, and the details on the figures are well-highlighted. Overall, the review article is very interesting, with up-to-date literature.

Minor revision:

  1. The numbering of a few sections needs to be changed for consistency. 3.3. should be 4.2 and so on.

Reply: The numbering has been corrected.

  1. The authors should explain how this review has improved from the previously published article on the same topic. (Molecular targets and anti-cancer potential of escin. Cancer Lett, 2018 May 28;422:1-8).

Reply: We thank the referee for mentioning this previous mini-review by Cheong et al. titled “Molecular targets and anti-cancer potential of escin” published in 2018. This mini-review indeed provides a good overview of anti-cancer potential of escin. However, our manuscript presents the most updated data on this topic since science always keep progressing. We provide a detailed overview of structures and chemical characteristics of escin – a feature that was not presented in any previous reviews focused on escin. Also, the current manuscript provides an overview of bioavailability and pharmacokinetics of escin. The previous review describes escin’s effects against 12 cancer types published by 22 studies. For comparison, our manuscript extends this information to 16 cancer types and 31 studies that clearly shows the topic to be of interest for scientific community.

  1. 3. The paper needs minor editing for English and grammatical errors.

Reply: We acknowledge the referee’s concerns for this. The manuscript has now been checked for errors and proofread by a native speaker.

Reviewer 2 Report

“Cancer diseases” is abrupt phrase kindly revisit it in the title of the manuscript.

Abstract is not too informative it seems as a part of introduction.

Introduction part is totally lacking about introduction of cancer its statistics and the relevance of newer natural phytocompounds as anticancer agents including other diseases. Kindly elaborate these facts substantially in the introduction part.

The introduction part is not in the line with the title of the manuscript. There is no glimpse of “multifaceted role” and “beyond cancer” is a very broad term which is again out of line with the present title.

The heading of the section 4 is wrong. There is nothing like “activities against cancer diseases”.

Moreover the subsections of this heading are again not in the line of the main heading as author is discussing anti-cancer, anti-inflammatory, antioxidant, anti-angiogenic.

The review article contains too little information on the said title.

The anticancer potential of the phytocompound has been beautifully revived by Cheong et al. 2018 in the manuscript titled “Molecular targets and anti-cancer potential of escin”. Thus how it is worth to publish the present manuscript?

Author failed to conclude the present study. The entire section 5 is comprised of future direction without quoting the present discussion appeared in the manuscript.

Overall the title, content (headings and sub headings) including conclusion and future direction required exhaustive rewriting, development of the content and restructuring of the manuscript.

Author Response

Reviewer 2:

  1. “Cancer diseases” is abrupt phrase kindly revisit it in the title of the manuscript.

Reply: The title has been revised now.

  1. Abstract is not too informative it seems as a part of introduction.

Reply: We thank the referee for their concerns to make our manuscript clearer and enjoyable to read. We now changed the abstract as following:

Although modern medicine is advancing at an unprecedented rate, basic challenges in cancer treatment and drug resistance remain. Exploiting natural product-based drugs is a strategy that has been proven over time to provide diverse and efficient approaches in patients care during treatment and post-treatment periods of various diseases, including cancer. Escin – a plant derived triterpenoid saponin – is one example of natural products with a broad therapeutic scope. Initially, escin was proven to manifest potent anti-inflammatory and anti-oedematous effects. But in the last two decades, other novel activities of escin that are relevant to cancer treatment have been reported. Recent studies demonstrated escin’s efficacy in compositions with other approved drugs to accomplish synergy and increased bioavailability to broaden apoptotic, anti-metastasis, and anti-angiogenetic effects. Here, we comprehensively discuss and present an overview of escin’s chemistry, bioavailability and highlight its biological activities against various cancer types. We conclude the review by presenting possible future directions of research involving escin for medical and pharmaceutical applications as well as for basic research.

  1. Introduction part is totally lacking about introduction of cancer its statistics and the relevance of newer natural phytocompounds as anticancer agents including other diseases. Kindly elaborate these facts substantially in the introduction part.

Reply: We thank the referee for their critical look at the introduction part. To the existing statistical information we added the following information now:

Cancer as a bottleneck of medicine is a leading death cause without differentiating wealthy and poor countries. Worldwide, it takes the lives of around 10 million people per annum [4]. This disease is characterised with abnormal growth of cells and results in very serious complications, poor quality of life and shortens the life expectancy. Cancer diagnosis and treatment is expensive and difficult, in fact, not always possible. Medicinal plants offer cheap, readily available sources of active pharmaceutical com-pounds (APCs) against many diseases. Approximately 25% of all the newly approved FDA drugs between 1981 and 2019 are considered to be natural products or derived from natural products [3].In this time frame, FDA approved 182 new drugs against cancer, 120 of which were natural product-based drugs [3].

  1. The introduction part is not in the line with the title of the manuscript. There is no glimpse of “multifaceted role” and “beyond cancer” is a very broad term which is again out of line with the present title.

Reply: We thank the referee for their comments about the title and introduction parts. The title has been changed and we modified the relevant paragraph of the introduction part as following:

One group of multifaceted APCs of plant origin is escin. These are triterpenoid saponins usually isolated from the seeds of Horse chestnut trees. As we show below, more than 30 escin isomers have been isolated and studied. Originally, escin was proven to be efficient against inflammation, chronic venous insufficiency and gastro-protection. But over the past two decades, numerous novel activities of escin were published such as anti-oxidative and anti-cancer effects. These studies demonstrated that escin has anti-proliferative and anti-metastasis effects against more than 15 different cancer types. In addition, a new trend – composite drugs of an escin and other APCs – is on the rise. Therefore, escin’s broad scope of biological activities combined with various composite drugs make it appealing in cancer treatment and post-treatment care.

  1. The heading of the section 4 is wrong. There is nothing like “activities against cancer diseases”.

Moreover the subsections of this heading are again not in the line of the main heading as author is discussing anti-cancer, anti-inflammatory, antioxidant, anti-angiogenic.

Reply: We thank the referee for concerns regarding section 4. The title of the section has now been changed.

  1. The review article contains too little information on the said title.

Reply: We appreciate referees comment and assume that they are indicating to the word “beyond” in the title. We have now changed the title accordingly. We can assure the referee that we have collected all the publication on the topic from reliable databases such as Web of Science Core Collection.

  1. The anticancer potential of the phytocompound has been beautifully revived by Cheong et al. 2018 in the manuscript titled “Molecular targets and anti-cancer potential of escin”. Thus how it is worth to publish the present manuscript?

Reply: We thank the referee for mentioning this previous mini-review by Cheong et al. titled “Molecular targets and anti-cancer potential of escin” published in 2018. This mini-review indeed provides a good overview of anti-cancer potential of escin. However, our manuscript presents the most updated data on this topic since science always keep progressing. We provide a detailed overview of structures and chemical characteristics of escin – a feature that was not presented in any previous reviews focused on escin. Also, the current manuscript provides an overview of bioavailability and pharmacokinetics of escin. The previous review describes escin’s effects against 12 cancer types published by 22 studies. For comparison, our manuscript extends this information to 16 cancer types and 31 studies that clearly shows the topic to be of interest for scientific community.

  1. Author failed to conclude the present study. The entire section 5 is comprised of future direction without quoting the present discussion appeared in the manuscript.

Reply: We have modified the section 5 as following:

In this paper, we provided a comprehensive discussion of escin’s chemistry, structural characteristics, and its multifaceted role in cancer therapy. Escin’s chemistry and pysico-chemical properties, including structural information of 34 isomers were presented. We showed that the there is ample evidence to suggest the anti-cancer ef-fects of escin to be mainly realised through the NF-κB pathway and can be summarised into 3 groups: apoptotic, anti-cell proliferative and anti-metastasis. Escin’s broad bio-logical activity scope enables it to be used as a potential anti-cancer compound, a sup-plementary medicine in cancer therapy and as an adjunct compound in composite drugs. In addition, escin can be used in treatment and post-treatment periods of cancer therapy due to its anti-inflammatory, anti-oxidative, protective and ameliorative ef-fects. We reviewed these effects of escin and concluded that escin’s anti-inflammatory effects are realised by the GC-like mechanism.

Escin has been an interesting and wide-reaching topic in the pharmaceutical in-dustry. Hence, it is a subject of systematic research. Still, some critical research is needed to further advance the state of the art. For instance, pharmacological studies of escin were performed in vitro and in vivo; however, as technology is progressing, there are now new methods and techniques that can offer more precise analysis. One such trend is single cell studies. As mechanisms of escin’s pharmacological activities are not yet fully resolved, single cell experiments can provide a platform to elucidate mecha-nisms of action. This is especially true for biological activities of escin against cancer cells. Computational advances allow us to work with big data, and high throughput single cell experiments are becoming popular. Therefore, it might be of high interest to reveal the finest mechanistic details of escin’s biological activities in the context of cancer. This should enable us to understand the nature of cancer diseases better and to increase the therapeutic efficacy of medicines, including escin against such diseases.

Another important area of research is to ensure protection of our natural re-sources. Development of new medicines based on escin’s structure can contribute to “green pharmacy”. Apart from escin’s use as an APC, it can be used  to develop drug carriers that are cheap, safe and with additional therapeutic effects. Modification of escin with synthetic or other biological surfactants is also an interesting direction. This could be an alternative way to design semi-synthetic surfactants with improved prop-erties that are crucial to produce drug delivery tools that increase solubility, bioavaila-bility and stability of drugs, especially those with high molecular weights. Such nano-particles and vesicles can enhance drug action by increasing the membrane’s permea-bility and sensitising cancer cells to chemotherapy drugs as well as by contributing to the therapeutic activity via its anti-inflammatory activity.

  1. Overall the title, content (headings and sub headings) including conclusion and future direction required exhaustive rewriting, development of the content and restructuring of the manuscript.

Reply: We thank the referee for their efforts to make this manuscript of high quality. The manuscript now been modified carefully.

Reviewer 3 Report

The manuscript entitled “Escin’s multifaceted roles in the treatment of cancer diseases and beyond: a comprehensive review” presents a review on chemistry, bioavailability and biological activities of escin against various cancer types.

The manuscript is quite interesting and well-written. However, minor revisions should be made in order to be published in Biomolecules journal, and the manuscript should be completed and/or modified taking into account the suggestions below:

1.     The authors are advised to rephrase the sentence  from lines 24-25, 74-75, 172-173, 248-249, 299-301.

2.     The authors are advised to extend the subsection 4.1. Anti-Cancer Effects

3.     The authors are advised to correct the subsection 3.3. Anti-Inflammatory Effects - 4.2, as well as the others subsections from section 4. Escin’s Biological Activities Against Cancer Diseases (4.3, 4.4)

4.     The authors are advised to present the references as indicated in Instructions for authors

Author Response

Reviewer 3.

The manuscript entitled “Escin’s multifaceted roles in the treatment of cancer diseases and beyond: a comprehensive review” presents a review on chemistry, bioavailability and biological activities of escin against various cancer types.

The manuscript is quite interesting and well-written. However, minor revisions should be made in order to be published in Biomolecules journal, and the manuscript should be completed and/or modified taking into account the suggestions below:

  1. The authors are advised to rephrase the sentence from lines 24-25, 74-75, 172-173, 248-249, 299-301.

Reply: We thank the referee for their concerns to make the manuscript clearer for the reader. The sentences have been revised and modified where required.

  1. The authors are advised to extend the subsection 4.1. Anti-Cancer Effects

Reply: We thank the referee for the comment. This subsection already contains all the publication that can be found on reliable databases such as Web of Science core collection or Scopus. We wrote this subsection in three subparts focusing on apoptotic, anti-cell proliferation and anti-metastasis activities of escin on various cancers. We wrote additional information for the reader to understand the molecular mechanisms of these activities. Escin’s antiangiogenic effects that are relevant to the cancer metastasis were given as a separate subsection. Since there are no other studies involving escin reporting its abovementioned activities it’s impossible to extend this subsection. We could delineate each study and describe how it’s done, but this would not fit for this journal and may drift the reader from the main points. We hope this review represents a big picture on the said topic and equips researchers with the most up to date information and directs to the original sources for further information.

  1. The authors are advised to correct the subsection 3.3. Anti-Inflammatory Effects - 4.2, as well as the others subsections from section 4. Escin’s Biological Activities Against Cancer Diseases (4.3, 4.4)

Reply: The order of subsections has been changed and numbering has been corrected now.

  1. The authors are advised to present the references as indicated in Instructions for authors.

Reply: We thank the referee for this advise. It has been corrected now.
